# The Gintonin-Enriched Fraction of Ginseng Regulates Lipid Metabolism and Browning via the cAMP-Protein Kinase a Signaling Pathway in Mice White Adipocytes

**DOI:** 10.3390/biom10071048

**Published:** 2020-07-15

**Authors:** Kippeum Lee, Heegu Jin, Sungwoo Chei, Hyun-Ji Oh, Sun-Hye Choi, Seung-Yeol Nah, Boo-Yong Lee

**Affiliations:** 1Department of Food Science and Biotechnology, College of Life Science, CHA University, Seongnam, Kyonggi-do 13488, Korea; joy4917@hanmail.net (K.L.); heegu94@hanmail.net (H.J.); sungwoochei@gmail.com (S.C.); guswl264@naver.com (H.-J.O.); 2Ginsentology Research Laboratory and Department of Physiology, College of Veterinary Medicine and BioMolecular Informatics Center, Konkuk University, Seoul 05030, Korea; vettman@naver.com (S.-H.C.); synah@konkuk.ac.kr (S.-Y.N.)

**Keywords:** ginseng, gintonin-enriched fraction, white adipocyte, triglycerides, browning, obesity

## Abstract

Obesity is a major health concern and is becoming an increasingly serious societal problem worldwide. The browning of white adipocytes has received considerable attention because of its potential protective effect against obesity-related metabolic disease. The gintonin-enriched fraction (GEF) is a non-saponin, glycolipoprotein component of ginseng that is known to have neuroprotective and anti-inflammatory effects. However, the anti-obesity and browning effects of GEF have not been explored to date. Therefore, we aimed to determine whether GEF has a preventive effect against obesity. We differentiated 3T3-L1 cells and mouse primary subcutaneous adipocytes for 8 days in the presence or absence of GEF, and then measured the expression of intermediates in signaling pathways that regulate triglyceride (TG) synthesis and browning by Western blotting and immunofluorescence analysis. We found that GEF reduced lipid accumulation by reducing the expression of pro-adipogenic and lipogenic factors, and increased lipolysis and thermogenesis, which may be mediated by an increase in the phosphorylation of protein kinase A. These findings suggest that GEF may induce fat metabolism and energy expenditure in white adipocytes and therefore may represent a potential treatment for obesity.

## 1. Introduction

Obesity is a major risk factor for type 2 diabetes, cardiovascular disease, hypertension, hyperlipidemia, and some forms of cancer [1]. It is characterized by an enlargement of adipose tissue depots to store excess energy in the form of triglycerides (TGs). There is an urgent need for effective anti-obesity therapies, and one approach that has been identified is to stimulate energy expenditure.

Adipose tissue plays a vital role in metabolic homeostasis and the regulation of energy balance [2]. In mammals, adipose tissue can be classified as white adipose tissue (WAT) and brown adipose tissue (BAT). WAT predominantly stores chemical energy as TGs, whereas BAT is specialized to dissipate energy by thermogenesis [3]. Recent studies have shown that cells with a BAT-like phenotype also exist in WAT, which are referred to as BAT-like or beige adipocytes. Therefore, numerous studies have investigated whether WAT-to-BAT transdifferentiation can be induced by treatment with chemical agents [4]. This “browning” of WAT is characterized by an increase in the expression of uncoupling protein 1 (UCP1) and mitochondrial expansion. UCP1 is a key BAT-specific thermogenic protein that generates heat by permitting significant proton leakage across the inner mitochondrial membrane [5]. UCP1 expression is characteristic of the BAT phenotype, and its expression is stimulated by transcription factors, including PR domain-containing 16 (PRDM16) and peroxisome proliferator-activated receptor gamma co-activator 1 α (PGC1α) [6]. Therefore, the stimulation of adipocyte transdifferentiation in WAT could represent a therapeutic strategy for obesity and obesity-related metabolic disease.

The cAMP-dependent protein kinase A (PKA) signaling pathway is an important regulator of energy metabolism in adipocytes [7]. Specifically, PKA signaling is important for adipogenesis, lipolysis, and mitochondrial biogenesis in adipocytes. Activation of the cAMP-dependent pathway, which involves the phosphorylation of PKA, stimulates lipolysis by activating both adipose triglyceride lipase (ATGL) and hormone-sensitive lipase (HSL), which hydrolyze TGs in the lipid droplets of white adipocytes [8,9]. The released free fatty acids can undergo mitochondrial β-oxidation [10], which generates acetyl CoA that is required for WAT browning. Indeed, a recent study showed that PKA activation upregulates UCP1 expression in beige adipocytes [11]. In addition, PKA inhibits intracellular TG accumulation and adipogenesis in white adipocytes [12]. Adipogenesis is the cellular differentiation process that transforms pre-adipocytes into mature adipocytes, and requires the expression of adipogenic factors, including CCAAT/enhancer-binding protein alpha (C/EBPα), peroxisome proliferator-activated receptor gamma (PPARγ), and fatty acid-binding protein 4 (FABP4) [13]. PKA also regulates the activity of lipogenic enzymes, such as lysophosphatidic acid acyltransferase theta (LPAATθ), diacylglycerol acyltransferase (DGAT), and phosphatidate phosphatase 1 (lipin1), which play important roles in the lipid accumulation that occurs during adipogenesis [14].

Korean ginseng (*Panax ginseng* Meyer, Araliaceae family) is a well-known medicinal herb that is used in Asian countries [15]. It has been used as a general tonic or adaptogen to increase the physical response to stress or fatigue, and to treat diseases such as cancer and diabetes mellitus [16,17]. Korean ginseng is reported to have numerous therapeutic effects that are mediated by its active components, which comprise saponins (referred to as “ginsenosides”), non-saponin components, phenolic ingredients, polysaccharides, and alkaloids [18,19,20]. Of these, ginsenosides have been the most intensively studied [21]. However, ginsenoside preparations are expensive because of their low concentrations in the plant and the complex process required for their isolation. Therefore, the biological activity of the non-saponin components of ginseng has also been investigated [22]. In a recent study, a novel glycolipoprotein fraction was isolated from ginseng, which was referred to as the “gintonin-enriched fraction” (GEF) [23]. Gintonin is composed of proteins that contain many hydrophobic and acidic amino acids along with glucose as a significant carbohydrate component [24]. In particular, according to a recent study, the major components of GEF are a complex of lysophosphatidic acids (LPA) and ginseng proteins including ginseng major latex-like protein151 (GLP151). GLP151 belongs to the plant Bet v 1 superfamily and represents the medicinal effect of GEF. Besides, it is reported that the GLP molecule is composed of 151 residues, and has the conserved helix–grip fold, which consists of three α-helices and a curved seven-stranded antiparallel β-sheet [25]. However, whether GEF has anti-obesity effects has yet to be determined. Therefore, in the present study, we determined the effects of GEF on fat metabolism, as well as the molecular mechanisms involved in 3T3-L1 and primary subcutaneous adipocytes.

## 2. Materials and Methods

### 2.1. Preparation of the Gintonin-Enriched Fraction

The GEF used in the present study was prepared as previously described [24]. Briefly, 4-year-old Korean white ginseng (Korea Ginseng Cooperation, Daejon, Korea) was chopped into small pieces (>3 mm) and refluxed with 70% ethanol for 8 h at 80 °C. The ethanolic extracts were then concentrated, dissolved in distilled water, precipitated, and lyophilized [26].

### 2.2. Cell Culture

Mouse 3T3-L1 pre-adipocytes (CL-173; American Type Culture Collection, Manassas, VA, USA) were cultured in Dulbecco’s modified Eagle’s medium (DMEM) containing 10% bovine calf serum (BS, Corning, NY, USA), 1% penicillin/streptomycin (P/S) solution, and 3.7 g/L sodium bicarbonate in a humidified 5% CO_2_ incubator at 37 °C. At 100% confluence, the cells were differentiated in DMEM containing 10% fetal bovine serum (FBS; Gibco, Gaithersburg, MD, USA), 10 µM dexamethasone, 0.5 mM 3-isobutyl-1-methylxanthine (IBMX), and 2 µg/mL insulin. After 2 days, the differentiation medium was replaced with maintenance medium (DMEM supplemented with 10% FBS and 5 mg/mL of insulin), which was refreshed every 2 days.

Mouse primary subcutaneous adipocytes (SAT) were obtained as described previously [27]. The stromal vascular fraction was isolated from the subcutaneous WAT of 5-week-old male ICR mice as follows. The ICR (CrljOri:CD1) mice were purchased from Joong-Ah Bio (Suwon, Korea). And this animal experiment was approved by the Institutional Animal Care and Use Committee (IACUC) of CHA University (IACUC approval number, 190173). Subcutaneous WAT was minced and digested in enzyme buffer (1.5 U/mL collagenase D, 2.4 U/mL Dispase II, and 10 mM CaCl_2_ in phosphate-buffered saline [PBS]), and then the digests were washed in PBS, and centrifuged at 1000× *g* for 15 min. The primary SATs obtained were incubated in Glutamax DMEM/F12 medium containing 10% FBS and 1% P/S until they reached confluence, when the medium was replaced with differentiation medium (DMEM supplemented with 10% FBS, 1% P/S, 100 µM indomethacin, 0.5 mM IBMX, 1 µM dexamethasone, and 5 µg/mL insulin) for 2 days. The differentiated SATs were maintained in DMEM containing 10% FBS, 1% P/S, and 5 µg/mL insulin.

GEF prepared in dimethyl sulfoxide was diluted with medium to 12, 25, or 50 µg/mL, and added to some of the cell cultures. To induce browning, 3T3-L1 cellss were cultured in differentiation medium supplemented with 10 nM triiodothyronine and 1 µM rosiglitazone. 7β-Acetoxy-8,13-epoxy-1α,6β,9α-trihydroxylabd-14-en-11-one (forskolin, 10 µM) or N-[2-(p-Bromocinnamylamino)ethyl]-5-isoquinolinesulfonamide dihydrochloride (H89, 10 µM) was also added to the differentiation medium of some of the cell cultures to determine the role of PKA.

### 2.3. Cell Viability Testing

Cells were seeded (4 × 10^3^ cells/well) in 96-well plates, GEF was added at 0, 6.25, 12.5, 25, 50, or 100 μg/mL, and the cells were incubated for a further 24 h. Then, 20 μL of 3-(4,5-dimethyl-2-thiazolyl)-2,5-diphenyl-2H-tetrazolium bromide (MTT) solution was added to each well. After a further 4-h incubation, the MTT-containing medium was removed and 100 μL DMSO was added to elute the formazan crystals. The absorbances of the eluates were measured at 570 nm using a Wallac 140 Victor 2 plate reader (Perkin-Elmer, Boston, MA, USA).

### 2.4. Oil Red O Staining

Differentiated cells were washed with PBS and fixed in 10% formaldehyde for 1 h at room temperature. After this, the cells were stained with 0.5% oil red O in a 60:40 *v/v* mixture of isopropanol and water for 1 h at room temperature. After washing and drying, the stained cells were imaged, and then the stain was eluted with 100% isopropanol, and the absorbances of the elutes were determined at 490 nm using a plate reader (BioTek Instruments Inc., Winooski, VT, USA).

### 2.5. Triglyceride Measurement

Differentiated cells were harvested in lysis buffer containing 1% Triton-100, 150 mM NaCl, 4 mM EDTA, 20 mM Tris-HCl (pH 7.4), and a protease inhibitor cocktail, and were lysed completely by sonication. The TG content, as an index of lipid accumulation, was quantitatively measured using a commercially available triglyceride assay kit (ZenBio, Research Triangle Park, NC, USA), according to the manufacturer’s instructions. The absorbances of the elutes were determined at 540 nm using the plate reader.

### 2.6. Western Blot Analysis

Cells were lysed in Pro-Prep solution (iNtRON Biotechnology, Seoul, Korea) containing phosphatase and protease inhibitors (Sigma-Aldrich, St. Louis, MO, USA). The lysates were clarified by centrifugation at 13,000× *g* for 150 min at 4 °C, and their protein concentrations were measured using a Bradford Assay (Bio-Rad, Hercules, CA, USA). The proteins in each sample (20 µg) were separated by sodium dodecyl sulphate-polyacrylamide gel electrophoresis and then transferred to polyvinylidene fluoride membranes. The membranes were blocked using 5% non-fat dried milk solution, and then immunoblotted with primary antibodies (1:1000) targeting carnitine palmitoyl transferase (CPT)1, PGC1α, PRDM16, UCP1 (all from Abcam, Cambridge, UK), C/EBPα, PPARγ, FABP4, ATGL, AMP-activated protein kinase (AMPK), p-AMPK (all from Cell Signaling Technology, Danvers, MA, USA), DGAT1, and glyceraldehyde 3-phosphate dehydrogenase (GAPDH) (both from Santa Cruz Biotechnology, Santa Cruz, CA, USA). The membranes were then incubated with horseradish peroxidase-conjugated secondary antibodies (Bio-Rad, Hercules, CA, USA) for 6 h (1:2000), and the reactive bands were detected using LAS image software (Fuji, New York, NY, USA).

### 2.7. Immunofluorescence Staining

Differentiated cells were seeded onto poly-L-lysine pre-treated cover slips (12 × 12 mm) and fixed using 4% paraformaldehyde for 20 min. For mitochondrial staining, 1 mM MitoTracker Red (Cell Signaling) was added before the fixation, according to the manufacturer’s protocol. The cells were blocked with 5% bovine serum albumin for 1 h at room temperature and then incubated with rabbit anti-UCP1 (1:500 dilution) antibody overnight at 4 °C. Alexa Fluor™ 594-conjugated and fluorescein isothiocyanate (FITC)-conjugated (1:1000 dilution) secondary antibodies were then applied, and the cell nuclei were stained using DAPI (Sigma-Aldrich). Finally, the cells were mounted using ProLong Gold Anti-fade reagent (Thermo Fisher Scientific, Waltham, MA, USA), and images were obtained using a Zeiss confocal laser scanning microscope (LSM880; Carl Zeiss, Oberkochen, Germany) and 2012 software (Carl Zeiss).

### 2.8. Statistical Analysis

All the data are presented as means ± standard deviations (SDs) and were the result of experiments conducted in at least triplicate. Significant differences were identified using one-way ANOVA and Duncan’s multiple range test in SAS 9.0 software (SAS Institute, Cary, NC, USA). Cell viability data were analyzed using Student’s *t*-test in SPSS software (IBM, Inc., Armonk, NY, USA).

## 3. Results

### 3.1. GEF Reduces Lipid Accumulation by Inhibiting Adipogenesis

We first determined the effect of GEF on fat metabolism in 3T3-L1s and primary SATs. The viability of both types of adipocytes following treatment with GEF was determined using an MTT assay. As shown in Figure 1A, 100 µg/mL GEF was cytotoxic; therefore, we used 12, 25, and 50 µg/mL concentrations in further experiments. To investigate the effect of GEF on fat accumulation in adipocytes, both types of pre-adipocytes were stimulated with IBMX, dexamethasone, and insulin, in the presence or absence of GEF for 8 days, and then oil red O staining was performed. As shown in Figure 1B,C, 25 and 50 µg/mL GEF significantly reduced lipid accumulation in 3T3-L1 cells (SAT imaging data not shown). In addition, we measured the expression of key transcription factors and biomarkers of adipocyte differentiation by Western blot analysis. GEF inhibited the expression of C/EBPα, PPARγ, and FABP4 in 3T3-L1s (Figure 1D,E) and SATs (Figure 1F,G). Specifically, treatment with 25 µg/mL GEF reduced the expression of C/EBPα by 65.3%, PPARγ by 43.5%, and FABP4 by 77.5% during the differentiation of 3T3-L1s, and it reduced the expression of C/EBPα by 86.1%, PPARγ by 40.8%, and FABP4 by 80.1% in SATs. These data indicate that GEF reduces fat accumulation during adipocyte differentiation, probably by regulating the expression of adipogenic genes.

### 3.2. GEF Reduces TG Synthesis, Probably by Regulating the Expression of Lipogenic Factors

To determine the effect of GEF on TG synthesis, we measured intracellular TG accumulation in 3T3-L1s and SATs. As shown in Figure 2A, lipid droplets were smaller in GEF-treated cells than in control of differentiation (CD). In regarding on it, we found that GEF reduced the TG content of both types of cells (Figure 2B,E). Numerous previous studies have shown that the expression of lipogenic factors, such as LPAATθ and DGAT1, is important for TG biosynthesis [28,29]. To obtain insight into the mechanisms underlying the TG-reducing effect of GEF, we next determined the effect of GEF on lipogenic protein expression in both types of cells and found that GEF reduced the expression of TG-biosynthetic enzymes, including LPAATθ and DGAT1. In particular, 25 µg/mL GEF reduced the expression of LPAATθ by up to 93.3% and 18.4% in 3T3-L1s and SATs, respectively. Consistent with this, GEF affected the morphology of both types of adipocytes, as shown in Figure 2G: the higher the concentration of GEF to which the adipocytes were exposed, the smaller their lipid droplets. These findings indicate that GEF may reduce TG synthesis in adipocytes by downregulating the expression of lipogenic factors.

### 3.3. GEF Increases the Phosphorylation of PKA and the Expression of Lipolytic Genes

Lipolysis in WAT liberates glycerol and free fatty acids from TGs, so that they can be used by other organs as energy substrates [30]. PKA phosphorylates target proteins, such as HSL and ATGL, which are key enzymes for TG hydrolysis, and for determining the balance between lipogenesis and lipolysis in adipocytes [31]. In the present study, we found that 25 µg/mL GEF significantly increased PKA phosphorylation in both types of white adipocytes. Treatment with 25 µg/mL GEF was also effective at increasing the expression of lipolytic genes (ATGL and phosphorylated-hormone sensitive lipase (p-HSL)) in 3T3-L1, but less effective in SATs (Figure 3A–D). In summary, GEF may reduce TG accumulation in adipocytes not only by inhibiting lipogenesis but also by inducing lipolysis.

### 3.4. GEF Upregulates the Expression of Thermogenic Transcription Factors

We next determined whether GEF induces 3T3-L1s and SATs to trans-differentiate to BAT-like adipocytes. CPT1 is a rate-limiting enzyme in mitochondrial fatty acid oxidation and is highly expressed in BAT, and fatty acid oxidation via CPT1 provides fuel for mitochondrial UCP1-mediated thermogenesis [32]. During browning, adipocytes start to express BAT-specific proteins, such as UCP1, which dissipates chemical energy in the form of heat [33]. In the present study, we found that GEF increased the expression of CPT1 in adipocytes in a dose-dependent manner, as shown in Figure 4. Moreover, GEF significantly increased the expression of PRDM16, PGC1α, and UCP1 in 3T3-L1s and SATs. Thus, GEF increases the expression of key mediators of adipose thermogenesis in white adipocytes, meaning that it may increase energy expenditure in these cells [34].

### 3.5. GEF Induces a Brown Fat-Like Phenotype in White Adipocytes

To evaluate the effect of GEF on thermogenesis, we compared 25 µg/mL GEF-treated cells with white adipocytes treated with well-known inducers of browning (50 nM triiodothyronine and 1 µM rosiglitazone). As shown in Figure 5A–D, 3T3-L1s and SATs treated with the browning inducers showed higher expression of p-PKA, CPT1, and thermogenic biomarkers (PGC1α, PRDM16 and UCP1). GEF further increased the expression of p-PKA and these proteins in cells treated with browning inducers. Furthermore, staining of both types of adipocytes with MitoTracker Red, and immunostaining with FITC-conjugated anti-UCP1 antibody, showed that the cytoplasmic staining intensities were increased by treatment with both the established inducers of browning and GEF (Figure 5E,F). Although there was no additive effect of thermogenic proteins, GEF treatment was as effective as browning inducer treatment. This implies that GEF increases mitochondrial activity and the expression of UCP1, which is consistent with a browning effect of GEF.

### 3.6. The Browning Effect of GEF Is Mediated via the Activation of PKA in White Adipocytes

cAMP and PKA are key components of the signaling pathway that activates adipogenesis, lipolysis, and WAT-to-BAT transdifferentiation [14,35]. To determine whether GEF increases UCP1 expression by activating PKA, a PKA activator (forskolin) or an inhibitor (H89) were added to pre-adipocytes during differentiation. Forskolin stimulates β-adrenergic signaling, and therefore lipolysis, by increasing the intracellular concentration of cAMP, which induces the PKA-dependent phosphorylation of HSL. By contrast, H89 is a well-known PKA inhibitor that significantly reduces basal PKA activity, and it was recently shown that H89 increases TG storage in adipocytes by reducing lipolysis [36]. On days 0–2 of culture, 3T3-L1 cells were treated or not with 10 µM H89 or 10 µM forskolin for 24 h, and were then stimulated to differentiate, after which they were analyzed. As shown in Figure 6, 10 µM forskolin up-regulated p-PKA expression, and GEF had an additive effect to that of forskolin in both cells. In addition, GEF increased p-HSL, CPT1 and CUP1 expression in both cells. In contrast, treatment of 10 μM H89 reduced the expression of p-PKA in both cells, but only SAT cells recovered protein levels by GEF. In contrast, the expression levels of CPT1 and UCP1 in H89-treated 3T3-L1 cells were slightly increased by GEF. The expression levels of p-HSL, CPT1 and UCP1 in H89-treated SAT cells were not significantly different by GEF. In summary, GEF may increase the expression of thermogenic genes in white adipocytes by activating PKA.

## 4. Discussion

Ginseng, the root of *Panax ginseng* Meyer, is a popular herbal medicine and functional health food that is consumed throughout the world [37]. In particular, Korean ginseng is known to have beneficial effects, including on energy levels, mood, and longevity [38]. In various studies, fresh ginseng takes 4 to 6 years to mature and is sourced from Asia. Additionally, white ginseng is peeled and dried under the sun either in its original state or after removing the outer layer [39,40]. In this study, we used GEF as a non-saponin in Korean white ginseng. This gintonin is extracted according to an established laboratory routine; this material has been studied for a long time, and the extraction routine is verified by a lot of peer reviewed studies [24]. The GEF used in this study was obtained from water fractionation after ethanol extraction from white ginseng, with 1.3% yield [41]. Recent studies have revealed that it has diverse effects on the nervous and other systems [42,43]. Gintonin is a non-saponin component of ginseng that consists of a group of glycolipoproteins with a mean molecular weight of about 67 kDa. There are at least six different forms of GEF, but all contain carbohydrates including glucose and glucosamine; hydrophobic amino acids such as linoleic, palmitic, oleic, and stearic acids; and a large amount of lysophosphatidic acid. Recent studies have shown that lysophosphatidic acid slows adipocyte differentiation and reduces the expression of PPARγ, and it plays important roles in lipid accumulation. [44,45]. However, the effects of GEF in adipocytes have yet to be determined. Therefore, we aimed to determine whether gintonin regulates lipid metabolism in white adipocytes, and therefore whether it might have an anti-obesity effect.

Classical white and brown adipocytes are derived from different cell lineages, but beige adipocytes have characteristics of both adipocytes [46]. The 3T3-L1 cell line is widely used to study white adipocyte differentiation in vitro, and SATs are widely used to interrogate the browning effects of biological substances [47]. The induction of transdifferentiation in white adipocytes is of particular interest to researchers interested in potential therapies for obesity. It has been shown that the expression of genes that are enriched in brown adipocytes can be induced in white adipocytes by catecholamine treatment, and that oxygen consumption can be increased in a UCP1-dependent manner [48]. Our previous study showed that ginsenoside Rg1, one of the saponin ingredients of ginseng, promotes thermogenesis by increasing UCP1 expression in 3T3-L1s [34]. In the present study, we aimed to determine whether GEF would also have a browning effect in white adipocytes, as well as the mechanism involved.

In the present study, we demonstrated that GEF, consisting of the glycolipoprotein component of ginseng, reduces lipid accumulation by reducing the expression of adipogenic proteins (C/EBPα, PPARγ, and FABP4) in 3T3-L1s and SATs. Furthermore, we showed that GEF significantly inhibits TG accumulation in these adipocytes in a dose-dependent fashion. Because TG comprises more than 90% of the volume of adipocytes, TG synthesis is a highly regulated pathway that is of utmost importance in white adipocytes. We also showed that GEF reduces the expression of lipogenic factors, including DGAT1, which is an enzyme that catalyzes TG biosynthesis and may also reduce the synthesis of TG by reducing LPAATθ expression, and thereby PPARγ expression.

Lipolysis is a catabolic process that mobilizes metabolic substrates in response to energy demand [49]. Lipolysis involves the hydrolysis of TG, which results in the release of fatty acids and glycerol, and requires the lipases, including ATGL and HSL. HSL is considered to be the key enzyme responsible for the hydrolysis of TGs stored in adipose tissue [50]. The phosphorylation of PKA in 3T3-L1s and SATs reduces fat accumulation by reducing the rate of TG hydrolysis, rather than by increasing TG synthesis. GEF treatment increased the phosphorylation of PKA and the expression of lipolytic enzymes (ATGL and p-HSL). Thus, GEF has the potential to reduce lipid accumulation by not only reducing adipogenesis and lipogenesis, but also by increasing lipolysis in white adipocytes.

The gene expression profile of 3T3-L1s and SATs is typical of white adipocytes, but β-adrenergic agonist treatment increases the expression of genes that are enriched in brown adipocytes and increases oxygen consumption in a UCP1-dependent manner. Therefore, numerous studies have investigated the effects of dietary ingredients on the browning of white adipocytes, and in particular whether they might have β-adrenergic agonist activity [51,52]. In the present study, we demonstrated that GEF increases WAT-to-BAT transdifferentiation during the differentiation of 3T3-L1s. Consistent with this possibility, GEF increased the expression of CPT1, PRDM16, PGC1α, and UCP1, which are key thermogenic markers, in white adipocytes. The thermogenic ability of brown adipocytes is conferred by UCP1, which is a proton transporter that uncouples electron transport from ATP production. In addition, PGC1α induces fatty acid and fat catabolism, and PRDM16 upregulates BAT-specific signaling in white adipocytes. Browning also involves mitochondrial biogenesis and an upregulation of fatty acid oxidation, which expends energy. We also found that GEF-treated 3T3-L1s demonstrate significantly higher expression of CPT1, which is the enzyme required for the transport of fatty acids from the cytoplasm into the mitochondria for oxidation, and therefore thermogenesis. Thus, GEF treatment enables energy derived from the oxidation of fatty acids to be dissipated as heat.

In the present study, we also compared the thermogenic effects of conventional inducers of browning triiodothyronine and rosiglitazone with that of GEF by measuring the expression of CPT1, PGC1α, and UCP1 following treatment of adipocytes with each or both of these. Triiodothyronine and rosiglitazone are well known to induce WAT-to-BAT transdifferentiation and energy expenditure by activating the β-adrenergic receptor pathway and inducing UCP1 expression [53]. However, we found that GEF-treated 3T3-L1s and SATs also showed higher thermogenic gene expression, including of UCP1, and mitochondrial activity. These findings imply that GEF increases mitochondrial mass, fatty acid oxidation, and UCP1 activity, which are all hallmarks of adipocyte browning.

Finally, we determined whether PKA phosphorylation is required for the effects of GEF on fatty acid oxidation and browning. In adipocytes, PKA activation increases lipolysis and mitochondrial activity by activating TG hydrolysis in lipid droplets. ATGL and p-HSL hydrolyze TG to monoacylglyceride, which is further hydrolyzed to glycerol and free fatty acids for oxidation in mitochondria. In the present study, GEF treatment of 3T3-L1s and SATs cultured with forskolin increased p-PKA expression. By contrast, there was no significant effect of GEF on the expression of p-PKA in GEF-treated 3T3-L1 cultured with H89. However, in case of SATs, GEF significantly recovered p-PKA reduction by H89 treatment. The expression of lipolysis and fatty acid oxidation genes (p-HSL and CTP1) were partially increased by GEF in H89-treated adipocytes. However, the expression of CTP1 and UCP1 was significantly increased by GEF in H89-treated 3T3-L1 cells. These data suggest that GEF induces browning in white adipocytes by increasing PKA phosphorylation, which upregulates lipolysis, fatty acid oxidation, and UCP1 expression. In summary, GEF may not only reduce fat accumulation but can also increase energy dissipation as heat, by upregulating UCP1 expression.

In conclusion, our data show that the activation of PKA by GEF inhibits fat accumulation and increases energy expenditure in white adipocytes. GEF inhibits both adipogenesis and lipogenesis, and increases the expression of lipolytic enzymes, thereby reducing triglyceride accumulation. GEF also increases the expression of CPT1, which is likely to result in greater fatty acid oxidation, and has additive effects to conventional inducers of browning, upregulating the expression of brown adipocyte-specific factors, including PRDM16 and UCP1. Thus, further study of GEF may yield a therapeutic plant compound that can reduce obesity and improve metabolic health.

## Figures and Tables

**Figure 1 biomolecules-10-01048-f001:**
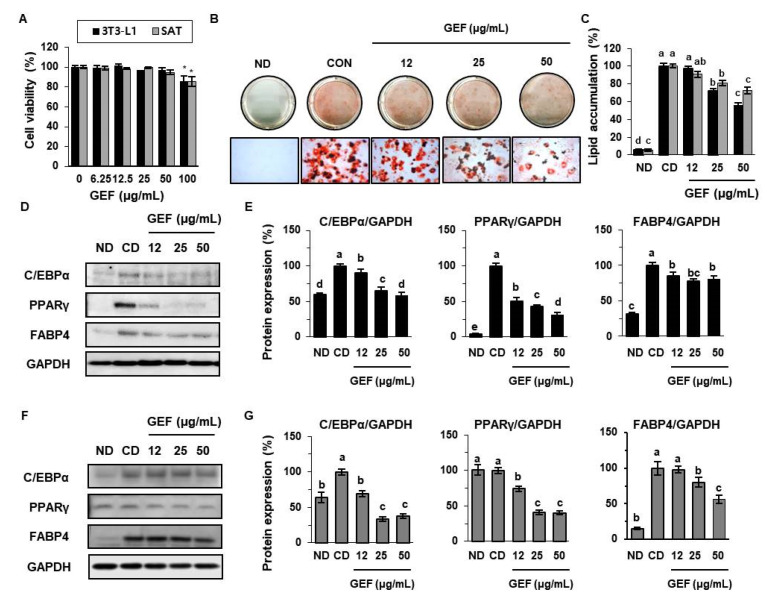
Effects of gintonin-enriched fraction (GEF) on lipid accumulation in 3T3-L1s and subcutaneous adipocytes (SATs). (**A**) Viability of 3T3-L1s and SATs treated with GEF for 24 h, determined using an MTT assay. (**B**,**C**) Photomicrographs and quantification of oil red O-stained adipocytes that had been treated with GEF during their 8 days of differentiation. (**D**,**E**) Expression of adipogenic proteins (CCAAT/enhancer-binding protein alpha (C/EBPα), peroxisome proliferator-activated receptor gamma (PPARγ), and fatty acid-binding protein 4 (FABP4)) in 3T3-L1s. (**F**,**G**) Expression of adipogenic proteins in SATs. Data are expressed as mean ± SD (*n* = 4). Treatments with different letters were significantly different, *p* < 0.05 (a > b > c > d > e). CD, control of differentiation; ND, undifferentiated.

**Figure 2 biomolecules-10-01048-f002:**
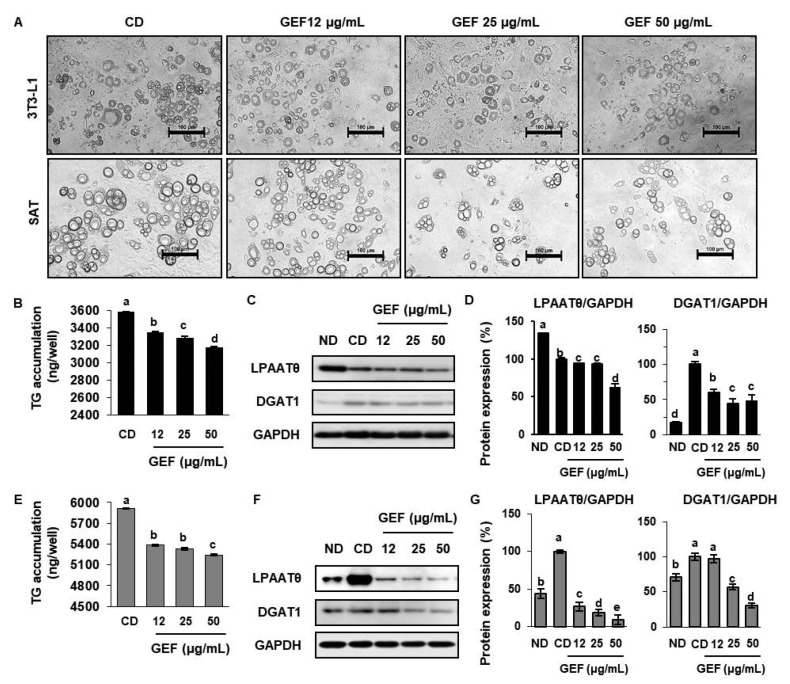
Effects of GEF on lipid droplet morphology and triglyceride synthesis in 3T3-L1s and SATs. (**A**) Cell morphology was evaluated by optical microscopy (400×). Adipocytes were differentiated for 16 days. Scale bar: 100 µm. (**B**,**E**) The intracellular triglyceride (TG) content of 3T3-L1s and SATs treated with GEF, measured using a TG ELISA kit. (**C**,**D**) Expression of lipogenic proteins (LPAATθ and DGAT1) in 3T3-L1s. (**F**,**G**) Expression of lipogenic proteins in SATs. Data are expressed as mean ± SD (*n* = 4). Treatments with different letters were significantly different, *p* < 0.05 (a > b > c > d).

**Figure 3 biomolecules-10-01048-f003:**
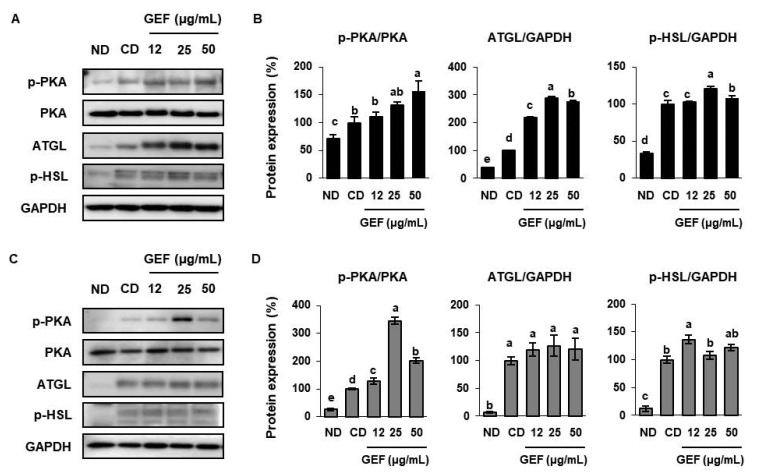
Effects of GEF on the phosphorylation of PKA, and expression of lipolytic enzymes in 3T3-L1s and SATs. (**A**,**B**) Expression of p-PKA/PKA and lipolytic enzymes (adipose triglyceride lipase (ATGL) and p-hormone-sensitive lipase (HSL)) in 3T3-L1s. (**C**,**D**) Expression of p-PKA/PKA and lipolytic enzymes in SATs. Treatments with different letters were significantly different, *p* < 0.05 (a > b > c > d > e).

**Figure 4 biomolecules-10-01048-f004:**
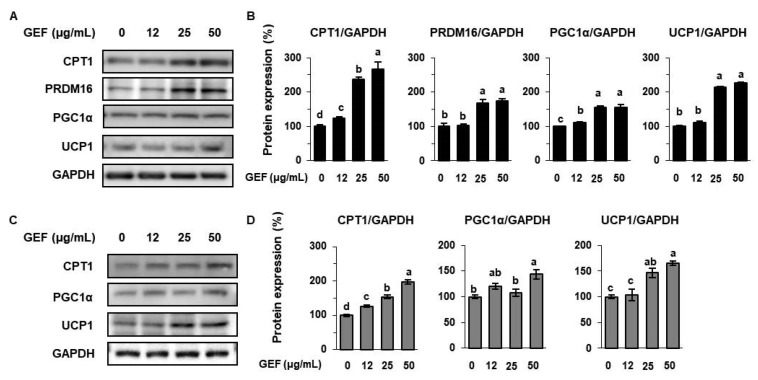
Effects of GEF on the expression of thermogenic proteins in 3T3-L1s and SATs. (**A**,**B**) Expression of thermogenic proteins (CPT1, PR domain-containing 16 (PRDM16), peroxisome proliferator-activated receptor gamma co-activator 1 α (PGC1α), and uncoupling protein 1 (UCP1)) in 3T3-L1s. (**C**,**D**) Expression of thermogenic proteins in SATs. Treatments with different letters were significantly different, *p* < 0.05 (a > b > c > d).

**Figure 5 biomolecules-10-01048-f005:**
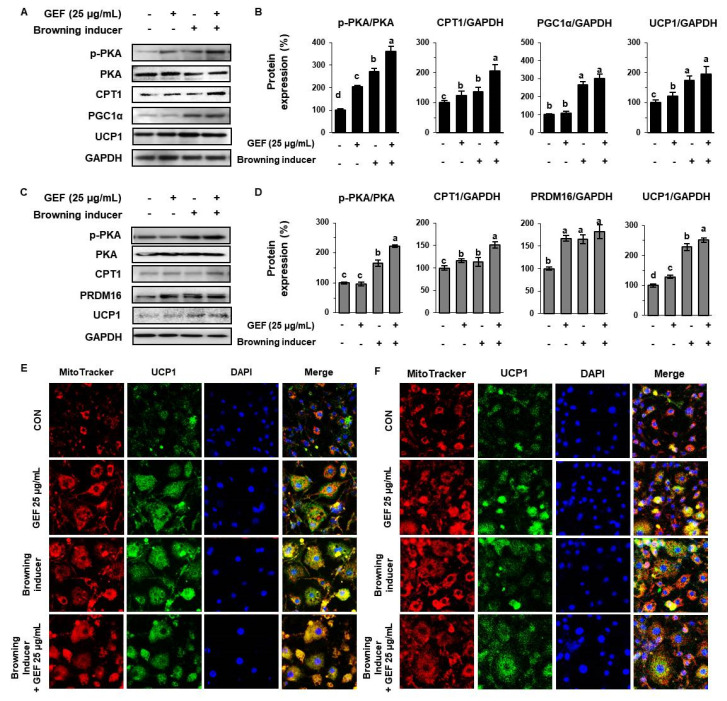
Additive effects of conventional browning inducers and GEF on thermogenesis in 3T3-L1s and SATs. (**A**,**B**) 3T3-L1s and (**C**,**D**) SATs were treated with 25 μM GEF with or without browning inducers (50 nM triiodothyronine and 1 mM rosiglitazone), and the expression of brown adipose tissue (BAT)-specific proteins was measured by Western blot analysis. (**E**,**F**) Immunofluorescence images of cells were captured at 800× magnification. Differentiated adipocytes were stained with MitoTracker Red and fixed with methanol, and then anti-UCP1 antibody and DAPI were applied (left, 3T3-L1s; right, SATs).

**Figure 6 biomolecules-10-01048-f006:**
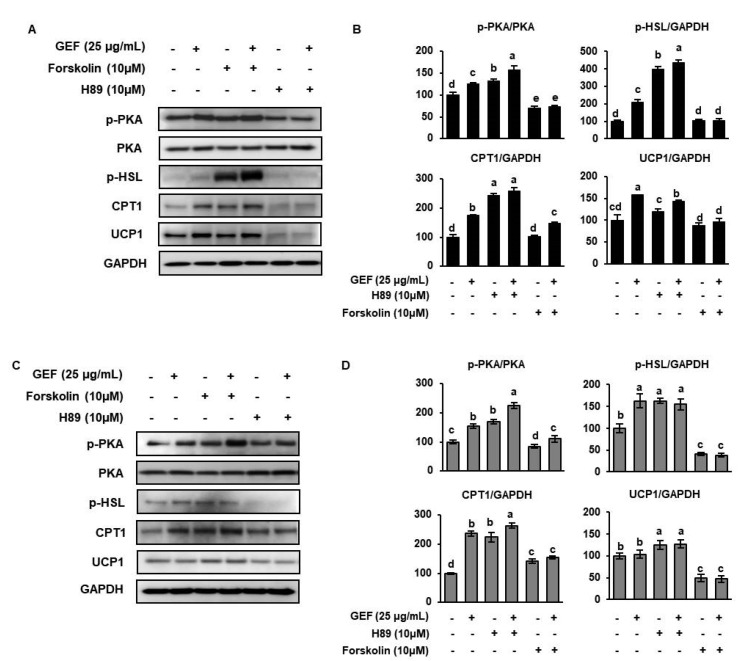
Effects of PKA activation and inhibition in 3T3-L1s and SATs treated with GEF. (**A**,**B**) 3T3-L1s were treated with 10 μM forskolin or 10 μM H89 ± 25 μM GEF, and the effects on protein phosphorylation and/or expression were determined by Western blotting. (**C**,**D**) SATs were treated with 10 μM CC ± 25 μM GEF, and the effects on protein phosphorylation and/or expression were determined by Western blotting. Data are expressed as mean ± SD (*n* = 4). Treatments with different letters were significantly different, *p* < 0.05 (a > b > c > d). Forskolin is 7β-Acetoxy-8,13-epoxy-1α,6β,9α-trihydroxylabd-14-en-11-one, and H89 is N-[2-(p-Bromocinnamylamino)ethyl]-5- isoquinolinesulfonamide dihydrochloride.

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
