# Peer review of "The Gintonin-Enriched Fraction of Ginseng Regulates Lipid Metabolism and Browning via the cAMP-Protein Kinase a Signaling Pathway in Mice White Adipocytes"

_biomolecules, 2020, doi:10.3390/biom10071048_

Round 1
Reviewer 1 Report
In this manuscript, the authors showed that gintonin-enriched fraction(GEF) reduced lipid accumulation, increased lipolysis and thermogenesis in 3T3-L1 cell and murine primary subcutaneous adipocytes. These results suggest the possibility of GEF as a potential treatment for obesity as the authors claimed.
- In material and methods, the authors descript that GFE was extracted from 4-years old Korea white ginseng. As claimed by Korea Ginseng Cooperation, 6-years old Korea ginseng is most effective. Is there any reason for the authors to claim for the choice of 4-years old Korea white ginseng?
- At line 174, authors claimed the cytotoxic dose of GEF is 100µg/ml, while the effective dose is 25 µg/ml and 50µg/ml. Do authors have some explanation about the narrow safe area of GEF?
- In this manuscript, authors used anti-GADPH as loading control antibody in western blotting studies. As descripted in many studies, application of insulin can upregulate GADPH expression in 3T3 adipocytes. For this experiment, GADPH is not a good choice as loading control antibody.
- Results 3.1, authors description of results is not match the figure 1.
- Result 3.2, authors description of results is not march the figure 2.
- Result 3.5, authors descripted the result of p-PKA expression, while there were no such data in figure 5. There are no descriptions of Figure 5 E and F, and other descriptions are not march the figure 5.
Author Response
Cover letter for revisions of Biomolecules-822587
Dear reviewers,
Thank you for considering our manuscript for publication in Biomolecules. We are very pleasure to have been given the opportunity to revise our manuscript, “The Gintonin-Enriched Fraction of Ginseng Regulates Lipid Metabolism and Browning via the cAMP-Protein Kinase A Signaling Pathway in Murine White Adipocytes”. We have addressed the reviewer’s comments point-by-point and made the necessary changes to the manuscript.
In major revision, we have revised the title as “The Gintonin-Enriched Fraction of Ginseng Regulates Lipid Metabolism and Browning via the cAMP-Protein Kinase A Signaling Pathway in Mice White Adipocytes”. Also we have addressed the reviewer’s comment on Gintonin-Enriched Fraction (GEF), and added Western blot data implying that GEF has a potential to treatment for obesity. Therefore, we revised description in result and discussion section, and highlighted the changes with track changes in the manuscript.
We hope that the manuscript is acceptable for publication in Biomolecules. We appreciate to consider this paper for publication in Biomolecules, and declare that authors of this work have no conflict of interests.
Sincerely yours,
Boo-Yong Lee, Ph.D.
(Please see the attachment)
Reviewer1
1. In material and methods, the authors descript that GFE was extracted from 4-years old Korea white ginseng. As claimed by Korea Ginseng Cooperation, 6-years old Korea ginseng is most effective. Is there any reason for the authors to claim for the choice of 4-years old Korea white ginseng?
(Answer) We appreciate your suggestion. As reviewer advised, we definitely agree that saponins of 6-years old Korea ginseng are most effective. However, fresh ginseng is a natural product that takes 4–6 years to mature and is sourced from Asia. Besides, white ginseng is produced by sun-drying or hot air-drying of 4–6-year-old fresh ginseng either in its original state or after removing the outer layer [1, 2]. We used GEF as a non-saponin in ginseng, from Dr.Nah’s laboratory. This gintonin is a laboratory routine to extract, this material has been studied for a long time, and it is verified by a lot of peer reviewed paper [3-6]. We appreciate your referencing the cited paper in adivance. We also added this description in Discussion.
2. At line 174, authors claimed the cytotoxic dose of GEF is 100µg/ml, while the effective dose is 25 µg/ml and 50µg/ml. Do authors have some explanation about the narrow safe area of GEF?
(Answer) Thank you for your kind suggestion. In our study, 100 µg/mL GEF have shown to cytotoxic in 3T3-L1 and SATs. Although the concentration of GEF (12-50 µg/mL) was narrow, we tried to investigate whether the concentration of GEF reduces fat accumulation while promoting browning.
3. In this manuscript, authors used anti-GADPH as loading control antibody in western blotting studies. As descripted in many studies, application of insulin can upregulate GADPH expression in 3T3 adipocytes. For this experiment, GADPH is not a good choice as loading control antibody.
(Answer) We agree that insulin can regulate GAPDH expression in 3T3-L1 adipocytes. However, in this study, insulin was used only for the purpose of differentiation induction for adipocytes, as an MDI induction substance with IBMX and Dexamethasone. According to reviewer’s comment, we checked the expression levels of β–actin instead of GAPDH using remained Western blots membrane. As a resultm the expression levels of GAPDH and β–actin have shown to same trend. Please refer to the WB data below. Therefore, we used GAPDH in this study. Please see the attachment
4. Results 3.1, authors description of results is not match the figure 1.
(Answer) Thank you for your kind advice. We revised Result 3.1 in page 5. Also, we investigated and changed data in Figure 1 FABP4 of SATs, by Western blot analysis.; “As shown in Figure 1B–C, 25 and 50 µg/mL GEF significantly reduced lipid accumulation in 3T3-L1 cells (SAT imaging data not shown). In addition, we measured the expression of key transcription factors and biomarkers of adipocyte differentiation by western blot analysis. GEF significantly inhibited the expression of C/EBPα, PPARγ, and FABP4 in 3T3-L1s (Figure 1D-E) and SATs (Figure 1F-G). Specifically, treatment with 25 µg/mL GEF reduced the expression of C/EBPα by 65.3%, PPARγ by 43.5 and FABP4 by 77.5% during the differentiation of 3T3-L1s, and it reduced the expression of C/EBPα by 86.1%, PPARγ by 40.8%, and FABP4 by 80.1% in SATs.” Please see the attachment.
5.Result 3.2, author’s description of results is not march the figure 2.
(Answer) Thank you for your advice. As reviewer said, GEF didn’t dose-dependently reduce the expression of LPAATθ and DGAT1. Only three concentrations were tested in the experiment and no significant difference between CD and 12 µg/mL GEF for LPAATθ and between 25 and 50 µg/mL GEF for DGAT1 in 3T3-L1s. A similar observation was shown, no significant difference between 25 and 50 µg/mL GEF for LPAATθ and between CD and 12 µg/mL for DGAT1 in SATs. Thus, we removed “doe-dependently reduced” in Result 3.2.; “To determine the effect of GEF on TG synthesis, we measured intracellular TG accumulation in 3T3-L1s and SATs. As shown in Figure 2A, lipid droplets were smaller in GEF-treated cells than in CD. In regarding on it, we found that GEF reduced the TG content of both types of cells (Figure 2B and E). Numerous previous studies have shown that the expression of lipogenic factors, such as LPAATθ and DGAT1, is important for TG biosynthesis [7, 8]. To obtain insight into the mechanisms underlying the TG-reducing effect of GEF, we next determined the effect of GEF on lipogenic protein expression in both types of cells and found that GEF reduced the expression of TG-biosynthetic enzymes, including LPAATθ and DGAT1. In particular, 25 µg/mL GEF reduced the expression of LPAATθ by up to 93.3% and 18.4% in 3T3-L1s and SATs, respectively.” Please see the attachment.
6. Result 3.5, authors descripted the result of p-PKA expression, while there were no such data in figure 5. There are no descriptions of Figure 5 E and F, and other descriptions are not march the figure 5.
(Answer) We appreciated for letting us know our mistake. We revised description and added p-PKA WB data in Figure 3.5.; “As shown in Figure 5A–B, 3T3-L1s and SATs treated with the browning inducers showed higher expression of p-PKA, CPT1, and thermogenic biomarkers (PGC1α, PRDM16 and UCP1). GEF further increased the expression of p-PKA and these proteinesin cells treated with these inducers. Furthermore, staining of both types of adipocytes with MitoTracker Red, and immunostaining with FITC-conjugated anti-UCP1 antibody, showed that the cytoplasmic staining intensities were increased by treatment with both the established inducers of browning and GEF (Figure 5E-F). Although there was no additive effect of thermogenic proteins, GEF treatment was as effective as browning inducer treatment.” Please see the attachment.

Reviewer 2 Report
Manuscript ID: Biomolecules-822587
Title: The Gintonin-Enriched Fraction of Ginseng Regulates Lipid Metabolism and Browning via the cAMP-Protein Kinase A Signaling Pathway in Murine White Adipocytes
Generally, this is an interesting topic to study the gintonin-enriched fraction of ginseng on lipid metabolism and browning in adipocytes. However, several descriptive mistakes need revisions. More precise description is suggested.
- Line 178-179, there was no significant difference when treated with 12 µg/mL.
- Line 181-182, Figure 1C and 1D should be 1E and 1G, respectively.
- Line 184, check the expression data in SATs. GEF non-significantly affected the levels of FABP4/GAPDH.
- Figure 1, a consistent symbol for control is required for the readers, con (in Figure 1B), CD (in Figure 1C-G) or CN (in the legend), and what CD stands for? Please clearly label.
- Line 199-202, GEF didn’t dose-dependently reduce the expression of LPAATθ and DGAT1 as the authors claimed. Only three concentrations were tested in the experiment and no significant difference between CD and 12 µg/mL GEF for LPAATθ and between 25 and 50 µg/mL GEF for DGAT1 in 3T3-L1s. A similar observation was shown, no significant difference between 25 and 50 µg/mL GEF for LPAATθ and between CD and 12 µg/mL for DGAT1 in SATs. Thus, it can’t conclude dose-dependent effect.
- Line 203-204, check the levels of LPAATθ expression for 3T3-L1 and SATs.
- Line 205, Check Figure 2 G or 2A?
- Line 220-222, the description was not identical well with the present data in Figure 3A-D.
- Line 222-223, check the data. The expression of lipolytic genes increased when treated with 25 µg/mL GEF was only shown in 3T3-L1s not in SATs.
- Line 269-279, the description was not identical well with the present data in Figure 6. Check the legend: (C-D) 3T3-L1s?
- Line 355-363, the description was not identical well with the present data in the manuscript.
Author Response
Cover letter for revisions of Biomolecules-822587
Dear reviewer,
Thank you for considering our manuscript for publication in Biomolecules. We are very pleasure to have been given the opportunity to revise our manuscript, “The Gintonin-Enriched Fraction of Ginseng Regulates Lipid Metabolism and Browning via the cAMP-Protein Kinase A Signaling Pathway in Murine White Adipocytes”. We have addressed the reviewer’s comments point-by-point and made the necessary changes to the manuscript.
In major revision, we have revised the title as “The Gintonin-Enriched Fraction of Ginseng Regulates Lipid Metabolism and Browning via the cAMP-Protein Kinase A Signaling Pathway in Mice White Adipocytes”. Also we have addressed the reviewer’s comment on Gintonin-Enriched Fraction (GEF), and added Western blot data implying that GEF has a potential to treatment for obesity. Therefore, we revised description in result and discussion section, and highlighted the changes with track changes in the manuscript.
We hope that the manuscript is acceptable for publication in Biomolecules. We appreciate to consider this paper for publication in Biomolecules, and declare that authors of this work have no conflict of interests.
Sincerely yours,
Boo-Yong Lee, Ph.D.
Please see the attachment
Reviewer2
1. Line 178-179, there was no significant difference when treated with 12 µg/mL.
(Answer) Thank you for your kind suggestion. We revised as “As shown in Figure 1B–C, 25 and 50 µg/mL GEF significantly reduced lipid accumulation in 3T3-L1 cells” in line 178-179.
2. Line 181-182, Figure 1C and 1D should be 1E and 1G, respectively.
(Answer) Thank you for your pointing out our mistake. We revised Figure 1 descriptions.
3. Line 184, check the expression data in SATs. GEF non-significantly affected the levels of FABP4/GAPDH.
(Answer) Thank you for your kind advice. We revised Result 3.1 in page 5. Also, we investigated and changed data in Figure 1 FABP4 of SATs, by Wstern blot analysis.; “As shown in Figure 1B–C, 25 and 50 µg/mL GEF significantly reduced lipid accumulation in 3T3-L1 cells (SAT imaging data not shown). In addition, we measured the expression of key transcription factors and biomarkers of adipocyte differentiation by western blot analysis. GEF significantly inhibited the expression of C/EBPα, PPARγ, and FABP4 in 3T3-L1s (Figure 1D-E) and SATs (Figure 1F-G). Specifically, treatment with 25 µg/mL GEF reduced the expression of C/EBPα by 65.3%, PPARγ by 43.5 and FABP4 by 77.5% during the differentiation of 3T3-L1s, and it reduced the expression of C/EBPα by 86.1%, PPARγ by 40.8%, and FABP4 by 80.1% in SATs.” Please see the attachment.
4. Figure 1, a consistent symbol for control is required for the readers, con (in Figure 1B), CD (in Figure 1C-G) or CN (in the legend), and what CD stands for? Please clearly label.
(Answer) Thank you for your pointing out our mistake. We revised as we wrote in Question 3 above.
5. Line 199-202, GEF didn’t dose-dependently reduce the expression of LPAATθ and DGAT1 as the authors claimed. Only three concentrations were tested in the experiment and no significant difference between CD and 12 µg/mL GEF for LPAATθ and between 25 and 50 µg/mL GEF for DGAT1 in 3T3-L1s. A similar observation was shown, no significant difference between 25 and 50 µg/mL GEF for LPAATθ and between CD and 12 µg/mL for DGAT1 in SATs. Thus, it can’t conclude dose-dependent effect.
(Answer) Thank you for your kind advice. We removed “dose-dependently” in Result 3.2.; “To obtain insight into the mechanisms underlying the TG-reducing effect of GEF, we next determined the effect of GEF on lipogenic protein expression in both types of cells and found that GEF reduced the expression of TG-biosynthetic enzymes, including LPAATθ and DGAT1.”
6. Line 203-204, check the levels of LPAATθ expression for 3T3-L1 and SATs.
(Answer) Thank you for your kind advice. We revised % value and Figure 3.2.; “In particular, 25 µg/mL GEF reduced the expression of LPAATθ by up to 89.2% and 18.4% in 3T3-L1s and SATs, respectively.” Please see the attachment.
7. Line 205, Check Figure 2 G or 2A?
(Answer) We revised description of Result 3.2.; “To determine the effect of GEF on TG synthesis, we measured intracellular TG accumulation in 3T3-L1s and SATs. As shown in Figure 2A, lipid droplets were smaller in GEF-treated cells than in CD. In regarding on it, we found that GEF reduced the TG content of both types of cells (Figure 2B and E).”
8. Line 220-222, the description was not identical well with the present data in Figure 3A-D.
(Answer) Thank you for your kind advice. We revised Result 3.3.; “In the present study, we found that 25 µg/mL GEF significantly increased PKA phosphorylation in both types of white adipocytes. Treatment with 25 µg/mL GEF was also effective at increasing the expression of lipolytic genes (ATGL and p-HSL) in 3T3-L1, but less effective in SATs (Figure 3A–D).”
9. Line 222-223, check the data. The expression of lipolytic genes increased when treated with 25 µg/mL GEF was only shown in 3T3-L1s not in SATs.
(Answer) We rewrote exactly Line 222-223.; Treatment with 25 µg/mL GEF was most effective at increasing the expression of lipolytic genes in 3T3-L1, but less effective in SATs.
10. Line 269-279, the description was not identical well with the present data in Figure 6. Check the legend: (C-D) 3T3-L1s?
(Answer) We revised Result 3.6 and its legend.; “As shown in Figure 6, 10 µM forskolin up-regulated p-PKA expression, and GEF had an additive effect to that of forskolin in both cells. In addition, GEF increased p-HSL, CPT1 and CUP1 expression in both cells. In contrast, treatment of 10 μM H89 reduced the expression of p-PKA in both cells, but only SAT cells recovered protein levels by GEF. In contrast, the expression levels of CPT1 and UCP1 in H89 treated 3T3-L1 cells were slightly increased by GEF. The expression levels of p-HSL, CPT1 and UCP1 in H89-treated SAT cells were not significantly different by GEF. In summary, GEF may increase the expression of thermogenic genes in white adipocytes by activating PKA.” Please see the attachment.
11. Line 355-363, the description was not identical well with the present data in the manuscript.
(Answer) Thank you for your comment. We revised Discussion.; “In the present study, GEF treatment of 3T3-L1s and SATs cultured with forskolin increased p-PKA expression. By contrast, there was no significant effect of GEF on the expression of p-PKA in GEF-treated 3T3-L1 cultured with H89. However, in case of SATs, GEF significantly recovered p-PKA reduction by H89 treatment. The expression of lipolysis and fatty acid oxidation genes (p-HSL and CTP1) were partially increased by GEF in H89-treated adipocytes."

Reviewer 3 Report
The authors have well explained at the cell level that it has a very significant effect on lipid metabolism and browning by Gintonin-enriched fraction. However, there are a few more things to consider about to augment the scientific meaning.
Major concerns
- The authors used MC3T3-L1 and the subcutaneous WAT of the ICR mice. Therefore, it would be better to take the term “mice” rather than the term “Murine” used in the title.
- It would be better to know the yield of GEF from the extraction method the authors used.
- In figure 2~6, it seems very important to GEF’s browning effect. If the subacute WAT had been separated from the mice, it would have been possible to take the Brown adipose tissue (BAT) and use it as a positive control. I may find negative control in this article, however, I cannot compare it to the amount of genes expressions that appear in the actual BAT of the mice.
- Triiodothyronine and rosiglitazone were compared with GEF. Why did you use triiodothyronine and rosiglitazone at the same time instead of doing it separately? If there is a special reason to mix the two agents together, please add an explanation.
- According to Inokuma et al., (2005), lipid metabolism-related to UCP1 explains that signals from AMP kinase and PI3 kinase are important. In response to this possibility, the authors used insulin in the badge used to grow cells, without any particular mention. I wonder what the WAT browning process you describe in terms of the energy efficiency used in mitochondria will have in comparison to the lipid metabolism presented in Inokuma et al., (2005).
(Inokuma K. et al., (2005) Uncoupling protein 1 is necessary for norepinephrine-induced glucose utilization in brown adipose tissue)
Minor concerns
- ‘CN’ should be modified to ‘CD’ in Figure 1 legends.
- A typo is found in Figure 5 legends. (1mM rosiglitazoFne)
Author Response
Cover letter for revisions of Biomolecules-822587
Dear reviewers,
Thank you for considering our manuscript for publication in Biomolecules. We are very pleasure to have been given the opportunity to revise our manuscript, “The Gintonin-Enriched Fraction of Ginseng Regulates Lipid Metabolism and Browning via the cAMP-Protein Kinase A Signaling Pathway in Murine White Adipocytes”. We have addressed the reviewer’s comments point-by-point and made the necessary changes to the manuscript.
In major revision, we have revised the title as “The Gintonin-Enriched Fraction of Ginseng Regulates Lipid Metabolism and Browning via the cAMP-Protein Kinase A Signaling Pathway in Mice White Adipocytes”. Also we have addressed the reviewer’s comment on Gintonin-Enriched Fraction (GEF), and added Western blot data implying that GEF has a potential to treatment for obesity. Therefore, we revised description in result and discussion section, and highlighted the changes with track changes in the manuscript.
We hope that the manuscript is acceptable for publication in Biomolecules. We appreciate to consider this paper for publication in Biomolecules, and declare that authors of this work have no conflict of interests.
Sincerely yours,
Boo-Yong Lee, Ph.D.
Please see the attachment
Reviewer3
1. The authors used MC3T3-L1 and the subcutaneous WAT of the ICR mice. Therefore, it would be better to take the term “mice” rather than the term “Murine” used in the title.
(Answer) As reviewer’s comment, we changed the title.
2. It would be better to know the yield of GEF from the extraction method the authors used.
(Answer) According to reviewer’s comments, we added ginseng information we used, in Discussion.; “In various studies, fresh ginseng takes 4 to 6 years to mature and is sourced from Asia. And white ginseng is peeled and dried under the sun with either in its original state or after removing the outer layer [1, 2].” In this study, we used GEF as a non-saponin in Korean white ginseng, from Dr.Nah’s laboratory. This gintonin is a laboratory routine to extract, this material has been studied for a long time, and it is verified by a lot of peer reviewed paper [3]. The GEF used in this study was obtained from water fractionation after ethanol extraction from white ginseng, with 1.3% yield [4].”
3. In figure 2~6, it seems very important to GEF’s browning effect. If the subacute WAT had been separated from the mice, it would have been possible to take the Brown adipose tissue (BAT) and use it as a positive control. I may find negative control in this article, however, I cannot compare it to the amount of genes expressions that appear in the actual BAT of the mice.
(Answer) Thank you for your suggestion. As the reviewer's advice suggests, the paper could be more convincing if we analyzed even the brown fat. We has been tried to investigate the BAT isolation as your advice. However, we could not BAT analyze as the positive control due to GEF material lack. The small amount of GEF we used in this study was produced in Dr.Nah's laboratory, so we appreciate your understanding in advance. However, fortunately, we have a schedule for next year's in vivo study using high-fat diet-induced male ICR mice with GEF oral administration. In that study, we can analyze anti-obesity effect via browning mechanism in mice WAT and BAT. Therefore, this in vitro study will be the background for future in vivo studies. In addition, many recent papers have investigated the effect of browning effect by treatment edible molecules in WAT cells [5, 6]. We designed and designed the research with reference to these papers.
4. Triiodothyronine and rosiglitazone were compared with GEF. Why did you use triiodothyronine and rosiglitazone at the same time instead of doing it separately? If there is a special reason to mix the two agents together, please add an explanation.
(Answer) To promote WAT browning, we tried to conduct various conditions of screening tests for determining the combination of browning inducers, 2 years ago. When triiodothyronine and rosiglitazone were used separately, browning marker UCP1 did not increase in our lab-conditions. After many try and failures, eventually, we referred method from Dr.Yun’s laboratory in Daegu University [5]. T3 is a thyroid hormone that not only regulates thermogenesis but also adjusts metabolism and energy balance. It activates brown fat thermogenesis by stimulating norepinephrine release and by increasing UCP1 gene expression [7]. Furthermore, recent studies demonstrated that T3 similarly promotes the browning of scWAT. In addition, Rsg, an antidiabetic PPARg ligand, has been shown to promote browning in WATs as a secondary effect, and 1 mM has been shown to be effective for this purpose in inguinal WATs [8]. To compare the browning effect of the well-known inducers T3 and Rsg with that of GEF, we therefore studied the expression levels of thermogenic genes.
Also, We have corrected and added Result 3.5 related with this.; “As shown in Figure 5A–B, 3T3-L1s and SATs treated with the browning inducers showed higher expression of p-PKA, CPT1, and thermogenic biomarkers (PGC1α, PRDM16 and UCP1). GEF further increased the expression of p-PKA and these proteinesin cells treated with these inducers. Furthermore, staining of both types of adipocytes with MitoTracker Red, and immunostaining with FITC-conjugated anti-UCP1 antibody, showed that the cytoplasmic staining intensities were increased by treatment with both the established inducers of browning and GEF (Figure 5E-F). Although there was no additive effect of thermogenic proteins, GEF treatment was as effective as browning inducer treatment.”
5.According to Inokuma et al., (2005), lipid metabolism-related to UCP1 explains that signals from AMP kinase and PI3 kinase are important. In response to this possibility, the authors used insulin in the badge used to grow cells, without any particular mention. I wonder what the WAT browning process you describe in terms of the energy efficiency used in mitochondria will have in comparison to the lipid metabolism presented in Inokuma et al., (2005).
(Inokuma K. et al., (2005) Uncoupling protein 1 is necessary for norepinephrine-induced glucose utilization in brown adipose tissue)
(Answer) Thank you for your kind advice. According to Inokuma et al., (2005), lipid metabolism-related to UCP1 can stimulate glucose utilization. Indeed, BAT can boost energy expenditure through uncoupled respiration mediated by UCP1 activation. And the possibility to use this ability to ameliorate obesity and diabetes has long been discussed [9]. Actually, as reviewer referred, we tried to investigate not only the anti-obesity effect through browsing, but also the effect of controlling blood sugar through increased GLUT4, at first. However, there was no effect on p-PI3K/PI3K and GLUT4 expression levels in 3T3-L1 by GEF treatment. Moreover, in this study, insulin was used for the purpose of differentiation induction for adipocytes, as an MDI induction substance with IBMX and Dexamethasone. Instead, we investigated the whether GEF affects on mitochondrial activity in both fat cells using immunofluorescence analysis, as shown in Figure 5E-F. As a result, treatment of GEF on adipocytes increased the staining level of MitoTracker Red, indicating that the activity of mitochondria increased. This mitochondrial activity can play the important role in increasing energy consumption during lipid metabolism in adipocytes [10]. Please see the attachment.
6. ‘CN’ should be modified to ‘CD’ in Figure 1 legends.
(Answer) Thank you for your pointing out our mistake. We revised Result 3.1 and legends. Also, we investigated and changed data in Figure 1 FABP4 of SATs, by Western blot analysis.; “As shown in Figure 1B–C, 25 and 50 µg/mL GEF significantly reduced lipid accumulation in 3T3-L1 cells (SAT imaging data not shown). In addition, we measured the expression of key transcription factors and biomarkers of adipocyte differentiation by western blot analysis. GEF significantly inhibited the expression of C/EBPα, PPARγ, and FABP4 in 3T3-L1s (Figure 1D-E) and SATs (Figure 1F-G). Specifically, treatment with 25 µg/mL GEF reduced the expression of C/EBPα by 65.3%, PPARγ by 43.5 and FABP4 by 77.5% during the differentiation of 3T3-L1s, and it reduced the expression of C/EBPα by 86.1%, PPARγ by 40.8%, and FABP4 by 80.1% in SATs.” Please see the attachment.
7. A typo is found in Figure 5 legends. (1mM rosiglitazoFne)
(Answer) Thank you for your pointing out our mistake. We revised as we wrote in Question 4 above

Reviewer 4 Report
Dear authors:
The work is attractive with a not very recurrent biological activity, the design is appropriate, however, strange the comparison of this fortified extract with a commercial anti-obesity drug to validate one more study.
On the other hand, I would ask you to include in the study the active principle and its structure of this enriching extract. And to improve all the figures especially the 1 and the 6.
Author Response
Cover letter for revisions of Biomolecules-822587
Dear reviewers,
Thank you for considering our manuscript for publication in Biomolecules. We are very pleasure to have been given the opportunity to revise our manuscript, “The Gintonin-Enriched Fraction of Ginseng Regulates Lipid Metabolism and Browning via the cAMP-Protein Kinase A Signaling Pathway in Murine White Adipocytes”. We have addressed the reviewer’s comments point-by-point and made the necessary changes to the manuscript.
In major revision, we have revised the title as “The Gintonin-Enriched Fraction of Ginseng Regulates Lipid Metabolism and Browning via the cAMP-Protein Kinase A Signaling Pathway in Mice White Adipocytes”. Also we have addressed the reviewer’s comment on Gintonin-Enriched Fraction (GEF), and added Western blot data implying that GEF has a potential to treatment for obesity. Therefore, we revised description in result and discussion section, and highlighted the changes with track changes in the manuscript.
We hope that the manuscript is acceptable for publication in Biomolecules. We appreciate to consider this paper for publication in Biomolecules, and declare that authors of this work have no conflict of interests.
Sincerely yours,
Boo-Yong Lee, Ph.D.
Please see the attachment
Reviewer4
The work is attractive with a not very recurrent biological activity, the design is appropriate, however, strange the comparison of this fortified extract with a commercial anti-obesity drug to validate one more study. On the other hand, I would ask you to include in the study the active principle and its structure of this enriching extract. And to improve all the figures especially the 1 and the 6.
(Answer) We appreciate for your kind advice.
First of all, according to reviewer’s comments, we added ginseng information we used, in Discussion.; “In various studies, fresh ginseng takes 4 to 6 years to mature and is sourced from Asia. And white ginseng is peeled and dried under the sun with either in its original state or after removing the outer layer [1, 2]. In this study, we used GEF as a non-saponin in Korean white ginseng, from Dr.Nah’s laboratory. This gintonin is a laboratory routine to extract, this material has been studied for a long time, and it is verified by a lot of peer reviewed paper [3].” The GEF used in this study was obtained from water fractionation after ethanol extraction from white ginseng, with 1.3% yield [4].”
Also, we revised Result 3.1.; “As shown in Figure 1B–C, 25 and 50 µg/mL GEF significantly reduced lipid accumulation in 3T3-L1 cells (SAT imaging data not shown). In addition, we measured the expression of key transcription factors and biomarkers of adipocyte differentiation by western blot analysis. GEF significantly inhibited the expression of C/EBPα, PPARγ, and FABP4 in 3T3-L1s (Figure 1D-E) and SATs (Figure 1F-G). Specifically, treatment with 25 µg/mL GEF reduced the expression of C/EBPα by 65.3%, PPARγ by 43.5 and FABP4 by 77.5% during the differentiation of 3T3-L1s, and it reduced the expression of C/EBPα by 86.1%, PPARγ by 40.8%, and FABP4 by 80.1% in SATs.”
Lastly, we also revised Result 3.6 and its legend.; “As shown in Figure 6, 10 µM forskolin up-regulated p-PKA expression, and GEF had an additive effect to that of forskolin in both cells. In addition, GEF increased p-HSL, CPT1 and CUP1 expression in both cells. In contrast, treatment of 10 μM H89 reduced the expression of p-PKA in both cells, but only SAT cells recovered protein levels by GEF. In contrast, the expression levels of CPT1 and UCP1 in H89 treated 3T3-L1 cells were slightly increased by GEF. The expression levels of p-HSL, CPT1 and UCP1 in H89-treated SAT cells were not significantly different by GEF. In summary, GEF may increase the expression of thermogenic genes in white adipocytes by activating PKA.”

Round 2
Reviewer 1 Report
The authors have answered all my questions and I basically agree with their revises. Therefore I think the revised manuscript could be accepted.
Reviewer 3 Report
The authors responded well to the reviewer's comments.